# Effect of Exposure of Plastic Infant Feeding Bottle Leached Water on Biochemical, Morphological and Oxidative Stress Parameters in Rats

**DOI:** 10.3390/toxics8020034

**Published:** 2020-05-13

**Authors:** Mahendra K. Pant, Abul H. Ahmad, Manisha Naithani, Hari S. Pandey, Monika Pandey, Jayanti Pant

**Affiliations:** 1Department of Anatomy, Government Doon Medical College, Dehradun, Uttarakhand 248001, India; pant.mahendra@gmail.com; 2Department of Veterinary Pharmacology & Toxicology, College of Veterinary & Animal Sciences, G.B. Pant University of Agriculture and Technology, Pantnagar, Uttarakhand 263145, India; ahahmadpharma@gmail.com; 3Department of Biochemistry, All India Institute of Medical Sciences, Rishikesh, Uttarakhand 249203, India; naithanimanisha@gmail.com; 4Department of Pathology, Government Doon Medical College, Dehradun, Uttarakhand 248001, India; physianin@yahoo.com; 5Department of Physiology, Government Medical College, Haldwani, Uttarakhand 263139, India; monikabiotch@gmail.com; 6Department of Physiology, All India Institute of Medical Sciences, Rishikesh, Uttarakhand 249203, India

**Keywords:** serum analysis, BPA, plastic leached water, oxidative stress, infant feeding bottles

## Abstract

Bisphenol A (BPA) is leached out from plastic infant feeding bottles that are filled with warm milk/water due to high temperatures, exposing the infants to BPA. The present study aims to understand the effects of ingestion of BPA leached from plastic infant feeding bottle and delineate the underlying mechanisms in rats. In this study, adult rats of Wistar strain were divided into 3 groups. In the first group, the rats consumed normal food and tap water *ad libitum*. In the second group, the rats ingested BPA (20 µg/kg bodyweight/day, orally). In the third group, the rats drank water leached from plastic infant feeding bottles. After 30days, tests involving biochemical parameters, histopathological examination, and oxidative stress enzyme markers were performed, and the levels of BPA in plastic-leached water were estimated by HPLC analysis. There were significant biochemical changes in the form of increased alkaline phosphatase (ALP), creatine kinase-muscle/brain (CK-MB), and lactate dehydrogenase (LDH) levels in both treated groups as compared to control group, accompanied by structural damage to the vital organs, and lipid peroxidation, glutathione reductase, and catalase activity were also high in the treated groups. Further, the BPA concentration in plastic leached water was estimated to be 0.1 ± 0.02 µg/mL.

## 1. Introduction

BisphenolA (BPA) is one of the chemicals with highest volume of production in the world. It is widely used in the manufacturing of plastic wares and also for the coating of thermal papers [1]. BPA is even used to line the inner walls of food and beverage cans to preserve edible substances [2,3]. BPA leaches out from plastic containers and food and beverage cans when they are exposed to high temperatures, acidic pH, or if cleansed or scrubbed using harsh detergents [4]. The leaching of BPA is reported to be higher from old worn plastic wares as compared to newer ones. Plastic wares are used enormously by people across the world. One of the common uses of plastic is in the form of infant feeding bottles, which are used by people at large. The easy availability, low cost, durability, and easy maintenance of plastic infant feeding bottles make them preferable for use by people in comparison to its glass counterpart, which requires much careful handling. Increased usage of plastics increases the risk of exposure to BPA. Studies have reported that the infant population is widely exposed to BPA [5]. This is marked by the presence of BPA in urine samples from infants [6]. Their exposure is often attributed to the presence of BPA in maternal milk or through top feeding [6]. Further, some studies have even reported leaching of BPA from infant bottles [4]. However, the toxic effects produced by consumption of BPA leached through these plastic bottles is little reported. Hence, in this study, we wanted to explore whether plastic infant bottles leach out BPA during routine use? What is the concentration of BPA leached out from these plastic infant feeding bottles during routine use? Does the consumption of BPA leached from these bottles produce any toxic effects on the organs and is the toxicity comparable with direct BPA ingestion? What are the underlying mechanisms of such toxicity?

BPA is reported to be an endocrine disruptor and acts on the estrogen receptors [7,8,9]. A wide number of studies have reported that it produces reproductive toxicity [10,11], decreased litter size per breeding pair, decreased fertility, and altered estrous cycles [12,13,14]. BPA is reported to affect the offspring survival and growth following maternal exposure at a dose >5 mg/kg bodyweight in rodents and young children [15,16]. BPA is also reported to cause neural and behavioral alterations in experimental animals [17,18,19,20]. Prenatal exposure of BPA is reported to be associated with childhood respiratory and allergic diseases [21]. This exposure produces hyperplasia of ducts of mammary glands and is implicated in inducing carcinogenesis of mammary glands, prostate, and others [22,23,24]. BPA is also reported to produce cardiac diseases, hepatic defects, and childhood obesity in a number of studies [25,26,27]. In our previous study, we reported the effects of acute and chronic exposure of BPA on cardiorespiratory parameters in rats [28,29]. Further, we also reported the effect of exposure of rats to plastic boiled water and its effect on phenylbiguanide-induced cardiorespiratory reflexes [30]. In this study, the rats were exposed to water obtained by boiling plastic bags in water. The toxic effects of leached BPA were revealed in this study; however, leached BPA obtained by routine use of plastic bottles was not studied.

Humans are exposed to BPA mainly through diet. After ingestion, BPA is conjugated in the liver to form BPA glucuronide and sulfate, which is excreted through the kidneys. The unconjugated BPA (free form) is biologically active, water-insoluble, and cannot be excreted by the kidneys. This free BPA remains inside body tissues. Further, infants have low levels of the enzymes needed to metabolize BPA [31].

As we carefully introspect into the method of using infant feeding bottles by people at large, we find that the plastic infant feeding bottles are initially cleansed by a detergent using a brush bottle cleaner. This is followed by rinsing off the detergent with tap water. Thereafter, the cleansed bottles are soaked into hot water for sterilization and allowed to dry. After drying, these bottles are filled with lukewarm or warm milk or water and used to feed the infants whenever required. In case the milk/water gets cold, these bottles are re-warmed by placing them in bottle warmers or keeping them in utensils filled with warm water. The likelihood of leaching of BPA increases many-fold in this entire process of cleaning and filling up the bottles with lukewarm/warm milk/water and further re-warming them in bottle warmers. Further, old bottles are more likely to leach BPA and cause further exposure of infants to the chemical.

Hence, we hypothesized that the milk/water ingested by the infants from plastic infant feeding bottles might contain leached BPA, which could possibly alter physiological mechanisms in the infants, since BPA is a known endocrine disruptor. Therefore, the present study was taken up to simulate a real life situation by feeding rats with water leached from plastic infant bottles, obtained after cleaning and filling the plastic infant feeding bottles in a way similar to that used for human infants, and examining the changes produced and further delineating the underlying mechanisms responsible for observed toxicity.

## 2. Materials and Methods

Adult female rats of Wistar strain (Weight 200–300 g; aged 10–12 weeks) were used in the present study. Ethical clearance from Institutional Animal Ethics Committee (IAEC/VPT/CVASc/281) was obtained to conduct the experiments. The present study was performed as a part of project funded by UCOST (Project sanction letter no.-UCS&T/R&D/MED-01/2015-16/9668; dated- 18 June 2015. 

### 2.1. Animal Maintenance

The rats were kept in an animal room at25 ± 0.5 °C, 50% RH, with 12:12 hlight/dark period. The animals were provided with food and water *ad libitum*. CPCSEA guidelines were followed while conducting the experiments.

The rats were divided into 3 groups. Each group comprised 6 rats. The sample size was calculated using the resource equation approach [32]. In Group 1 (control group), the rats were provided with food and RO-filtered tap water *ad libitum.*In Group 2 (BPA treated), BPA dissolved in olive oil (20 µg/kg bodyweight/day) was fed to the rats, orally, by gavage feeding.In Group 3 (plastic leached water treated), water leached from plastic infant bottles as mentioned under section “Drugs and Solutions” was provided to the rats. The rats consumed plastic leached water instead of RO-filtered tap water throughout the period of treatment. Each rat drank 30–35 mL of this leached water per day.

The animals were maintained in the above mentioned treatment for 30 days in all the groups.

### 2.2. Drugs and Solutions

BPA (analytical grade) was procured from Sigma Aldrich (USA). BPA solution was prepared by initially dissolving BPA in ethanol (Alpha chemika, Mumbai, India). Thereafter, it was diluted with olive oil (Figaro, Alcolea, Spain) to prepare the solution as per dosage (20 µg/kg bodyweight/day), which was fed to rats of Group 2.

Plastic leached water was prepared by following method: The infant feeding bottles used in the present study were made of plastic. These bottles were purchased from a local shop. The information regarding the type of plastic used in these bottles was not mentioned.Initially, the plastic infant bottles were cleansed with detergent and washed under tap water.These bottles were soaked in hot water for sterilization.Thereafter, the bottles were allowed to dry.The dried bottles were filled with water, which was previously boiled to 100 °C in a separate glass container. The water was boiled to 100 °C, as in real practice, water/milk is initially boiled by people for sterilization and, thereafter, it was allowed to cool so that it can be administered to infants, where it attains a lukewarm temperature.These filled infant bottles were kept in a water bath with the temperature maintained at 40 °C for an hour, so that lukewarm temperature was maintained.This water was given to the rats of Group 3.The leached water thus prepared was subjected to HPLC analysis to determine the concentration of BPA.

### 2.3. HPLC Analysis

HPLC analysis for BPA estimation in plastic leached water was performed using Shimadzu Liquid Chromatography System (Shimadzu Corporation, Kyoto, Japan, Model SPD-10A, LC10AT). The system comprised a double plunger pump, Rheodyne injector with 20 µL loop, and UV–VIS detector. Bisphenol A was separated by using C18 reverse phase column (Lichrospher 100 RP, 18.5µm (125 mm × 4 mm). The standard solutions of BPA were prepared by dissolving 2 mg of pure BPA in 2 mL of acetonitrile. Thereafter, various dilutions of 10, 5, 2.0, 1, 0.5, 0.2, 0.1, 0.05, 0.025, 0.01 µg/mL were prepared. A 20 µL aliquot of each of these prepared solutions was injected into HPLC. The mobile phase comprised water/acetonitrile (30:70 (*v*/*v*)) and the pH was adjusted to 4.3 with 0.1N HCl, with a flow rate of 0.5 mL, minus the isocratic form. The detector wavelength was set at 277 nm. A retention time of 3.3 min was recorded for Bisphenol A.

### 2.4. Histological Examination and Biochemical Parameter

After 30days, blood was withdrawn from the rats of all three groups. These blood samples were then centrifuged, and serum was separated. The serum thus obtained was used for biochemical analysis to study alkaline phosphatase (ALP), lactate dehydrogenase (LDH), serum glutamic oxaloacetic transaminase (SGOT), serum glutamic pyruvic transaminase (SGPT), bilirubin levels, very-low-density lipoprotein (VLDL), high-density lipoproteins (HDL), total cholesterol (TC), creatine kinase-muscle/brain (CK-MB), and blood urea levels.

Estimation of serum SGPT, SGOT, and ALP were done using the diagnostic reagent kit by DiaSys international. The plasma concentrations of TC, TG, and HDL-cholesterol were measured using spectrophotometric methods. Laboratory kit reagents (Randox Laboratory Ltd., UK) were used for urea and all lipid biochemical analysis and their absorbances were read using a Randox RX MISANO semi-automated clinical chemistry analyzer. Diagnostic kits for creatine kinase (CK-MB) isoenzyme and LDH were purchased from Accurex Biomedical Pvt. Ltd. (Gujarat, India).

Thereafter, the animals in all the groups were sacrificed and vital organs such as the liver, kidneys, and lungs were extracted and subjected to histological processing.

The tissue specimens were initially preserved in 10% neutral buffered formalin after excision. Thereafter, they were dehydrated using graded concentrations of ethanol. For this, the tissues were immersed in 70% ethanol in water, followed by 95% and 100% solutions. This was followed by clearing, in which dehydrating agent was replaced by xylene so that the tissue acquired a translucent appearance. Further, the tissues were impregnated with paraffin wax so as to prevent distortion of the tissue structure during microtomy. These steps were performed using an automated tissue processor (YorCo) following an overnight processing schedule.The processed tissue was thereafter subjected to microtomy and 5µm ultrathin sections were prepared using semiautomatic rotary microtome (Therma). These sections were floated on a thermostatically controlled water bath followed by placing over the hot plate for drying.

Finally, these tissue sections were transferred to 75 mm × 25 mm clean glass slides and were stained by hematoxylin and eosin (H&E) stain. For H&E staining, the sections were dewaxed and rehydrated through descending grades of alcohol to water. Fixation pigments were removed and stained in Harris hematoxylin for 5 min. This was followed by washing under running tap water until sections were ‘blue’. Then, they were differentiated in 1% acid alcohol for 30 s. They were again washed under running tap water until sections were ‘blue’. This was further followed by dipping in ammonia water, then by a 5 min tap water wash. Thereafter, they were stained in 1% eosin Y for 3 min followed by washing under running tap water for 2 min. They were dehydrated using graded concentrations of alcohol, then cleared and mounted. Once the staining process was complete, the sections were observed under binocular compound microscope for histopathological examination.

### 2.5. Oxidative Stress Markers

In addition to biochemical and histopathological examination, oxidative stress enzyme activity was also examined in the erythrocytes, liver, and kidneys of rats in all three groups. Absorbance was recorded using a UV–VIS spectrophotometer.

Lipid peroxidation (LPO) in the tissue samples was performed using a thiobarbituric acid-reactive substances (TBARS) test and formation of malondialdehyde (MDA) was examined [33]. Briefly, the homogenate samples were centrifuged at 736× *g* for 10 min and butylated hydroxyl toluene (BHT) (Sigma Aldrich, USA), TBA (Sigma Aldrich, USA), and HCl ((Fisher chemicals, India) were mixed with the supernatant. This mixture placed in a water bath at 95 °C for 10 min and was re-centrifuged. The absorbance of the supernatant thus obtained was read at 535 nm.

LPO in the erythrocytes was evaluated after separating the erythrocytes by centrifuging the heparinized blood and washing them in 0.15 M NaCl (Sigma Aldrich, USA) and diluting in PBS. The packed RBC was used for studying LPO using method of Shafiq-ur-Rehman [34].

The Goldberg and Spooner method was used to estimate glutathione reductase activity [35], while Aebi’s method was used to measure catalase activity [36].

### 2.6. Statistical Analysis

The data from all the three groups for all the parameters were analyzed. Statistical tests in the form of Mann–Whitney U test was applied. Values of *p* < 0.05 were taken as statistically significant.

## 3. Results

### 3.1. Biochemical Changes

The exposure of the rats to BPA and plastic leached water produced biochemical changes. The ALP levels were significantly high in both BPA-treated and plastic leached water-treated groups as compared to the control rats (Figure 1a; *p* < 0.05, Mann–Whitney U test). The LDH levels significantly increased in both BPA and plastic leached water-treated groups as compared to control group (Figure 1a; *p* < 0.05, Mann–Whitney U test). However, the SGOT levels were significantly reduced in the BPA-treated group as compared to control group (Figure 1b; *p* < 0.05, Mann–Whitney U test). Further, there was no change in the bilirubin levels (direct and indirect) in any of the groups.

Total cholesterol levels decreased significantly in BPA-treated and plastic leached water-treated groups respectively (Figure 1b; *p* < 0.05, Mann–Whitney U test). VLDL was reduced in the treated groups but the decrease was not significant. HDL values were increased in both treated groups; however, the change was not significant. CK-MB levels were increased significantly in both BPA-treated and plastic leached water-treated groups (Figure 1c); *p* < 0.05, Mann–Whitney U test). The urea level was significantly lowered in the plastic leached water-treated group as compared to the control group, however the BPA-treated group did not show any significant alterations (Figure 1c; *p* < 0.05, Mann–Whitney U test).

### 3.2. Cytoarchitectural Changes

Kidneys: The kidneys in the control group revealed healthy glomeruli and tubules with normal periglomerular space. Hyperplastic glomeruli were seen in both BPA-treated and plastic leached water-treated groups as compared to control group (Figure 2a,b). The hyperplastic changes are due to loss of number of glomeruli so that the remaining glomeruli have increased in size to compensate for the loss. This is seen as reduced periglomerular space in the higher magnification (Figure 2b). There is a loss of cellular details of tubular epithelial cells with focal cloudy swelling. The interstitium shows increased number of mononuclear inflammatory cell infiltrate in both treated groups as compared to control group. At places, extravasation of RBC seen in both the treated groups but more pronounced in BPA-treated groups (Figure 2a,b).

Liver: Liver sections from control group rats showed central veins with healthy hepatocytes. Distortion of lobular hepatic architecture with moderate mononuclear parenchymal infiltration with visible dilatation of central veins was seen in BPA-treated rats. In plastic leached water-treated group, there was a distortion of lobular hepatic sinusoids along with the central vein in the form of focal sinusoidal dilatation. Mononuclear cell infiltration in parenchyma was also seen in plastic leached water-treated group (Figure 3). In BPA-treated sections, foci of necrosis were also noted.

Lungs: The lungs obtained from control group rats showed clear alveoli with healthy lung parenchyma. Emphysematous changes were seen in both the treated groups. Areas of coalescence of air spaces resulting in to large air spaces were visible throughout the field in both BPA-treated and plastic leached water-treated group as compared to lungs of control group. Plastic leached water-treated group showed more prominent changes with much larger airspaces. Distortion of alveoli was seen in both the treated groups. The lung parenchyma appeared compressed, thinned out, and damaged in both the treated groups. The intervening lung parenchyma shows mild to moderate mononuclear inflammatory infiltrates with dysfunction of alveolar septae and collapse of total architecture. The emphysematous changes were more prominent in plastic leached water-treated group. Dilatation of alveoli was visible in both groups (Figure 4).

### 3.3. Oxidative Stress Tests

There was a significant increase in lipid peroxidation, glutathione reductase activity, and catalase activity in the erythrocytes, liver, and kidneys of rats of both BPA-treated and plastic leached water-treated groups as compared to the control group (Figure 5a–c).

### 3.4. BPA Estimation in Plastic Leached Water by HPLC

The concentration of BPA estimated in plastic leached water was estimated to be 0.1 ± 0.02 µg/mL by HPLC.

## 4. Discussion

The present study revealed that ingestion of BPA and plastic leached water by rats produced damage to vital organs which was seen in the form of biochemical, histological, and oxidative stress enzyme activity changes.

The biochemical tests showed increased levels of ALP, LDH, and CK-MB, whereas levels of SGOT, urea, and total cholesterol levels were decreased in the treated groups as compared to the control group. In a state of health, ALP is produced by liver, bones, intestine, pancreas, and kidneys [37], whereas LDH is mainly synthesized in liver, heart, pancreas, kidneys, skeletal muscles, and blood cells [38] and CK-MB is generated from the myocardium [39]. Increases in the levels of these enzymes are markers of cellular damage in these organs. SGOT is found in liver, heart, kidneys, skeletal muscles, brain, and red blood cells. Decreases in SGOT levels have been reported in the later course of ischemia, leading to hepatocyte injury [38]. Decreased SGOT levels in BPA-treated rats in the present study may be a representation of a chronic phase of ischemic injury of hepatocytes, such that less SGOT is produced. Further, low urea levels were observed in the plastic leached water-treated group. Low urea levels are indicative of liver disease, malnutrition, or overhydration [40]. In the present experimental setup, malnutrition or overhydration of the rats are a rare possibility, as they were under daily observation and treatment, further supporting liver injury as a possible cause of low urea levels, since improper functioning of liver leads to poor ammonia metabolism and less urea formation. Further, total cholesterol levels were significantly decreased in both the treated groups. BPA is a known endocrine-disrupting chemical and acts on the estrogenic receptors. BPA competes with estrogen and alters the lipid profile due to oxidative stress-induced injury, as mentioned in other studies [41]. We expected an increase in the cholesterol levels; however, the results of the present study showed decreased levels in treated groups along with insignificant changes in VLDL, HDL. The possible reason behind this kind of response may be attributed to low BPA doses, which were insufficient to produce the changes, or that exposure for 30 days was not enough to affect the lipid profile in treated rats, and chronic exposure could have altered the profile as reported in other studies, where duration of exposure to BPA was 8 weeks or the doses were much higher [41,42,43].

The biochemical changes seen in present study were substantiated by the histopathological examination of the tissues and oxidative stress enzyme activity results. The histopathology results showed prominent damage to the liver cytoarchitecture in the form of dilated central veins and mononuclear cell infiltration. Further, presence of glomerular hyperplasia along with inflammatory changes in the form of mononuclear cell infiltration and extravasation of erythrocytes and thinning of tubules were seen in the kidneys of rats of both the treated groups. Distinctive emphysematous changes were found in the lungs of rats of both the treated groups, especially in plastic leached water-treated rats, where there was widespread damage of the lung parenchyma.

Oxidative stress markers in the form of increased lipid peroxidation, glutathione reductase activity, and catalase enzyme activity in erythrocytes, liver, and kidney tissues in the treated groups helped in further delineating the mechanism of biochemical and morphological changes produced in the treated groups. High lipid peroxidation is linked to the state where the balance of oxidative damage and body’s antioxidant repair capacity is altered, with a shift towards oxidative damage leading to apoptosis or necrosis and cell death [44,45]. Glutathione is used in the electron transport system in the liver, and its reduction occurs in the presence of enzyme glutathione reductase [46,47]. Glutathione reductase is an important molecule to prevent oxidative stress and maintains the reduced state of cells. Increased activity of the enzyme in the treated groups indicates increased oxidative stress in tissues. Catalase is also another antioxidant enzyme which converts H_2_O_2_ into water and oxygen, and provides protection against oxidative stress. Increase in catalase activity in the present study may be correlated with increased oxidative stress load on the rats of the treated groups [48,49].

Our results showed that plastic leached water from the infant feeding bottles produced changes similar to direct ingestion of low dose of BPA (20 µg/kg bodyweight/day). The concentration of BPA in the plastic leached water was estimated to be 0.1 ± 0.02 µg/mL, which was, again, a very low dose of BPA. Plastic wares leach out numerous chemicals such as phthalates, BPS, and BPF in addition to BPA. These chemicals also possess estrogenic activity; however, detection of these chemicals was beyond the scope of the present study. However, it must be agreed, at this point, that some of these chemicals might have also contributed to the damage produced in the animals treated with plastic leached water along with leached BPA.

Looking into the metabolism of BPA, it is clear that after oral ingestion, BPA reaches the liver for conjugation. During the process of conjugation, insoluble free BPA is converted into the water-soluble conjugated form, which is excreted through the kidneys. The proposed mechanism of tissue damage in the present study is that exposure of the rats to direct and leached BPA for 30 days had built up oxidative stress in tissues with time, as seen in the results of oxidative stress enzyme activity. This might have induced damage of hepatocytes, as mentioned in results, in the form of deranged ALP, LDH, urea, and SGOT levels along with cytoarchitectural changes in the liver. Hepatocyte injury must have interfered with the BPA-metabolizing ability of liver and increased the levels of the water-insoluble free form of BPA (unconjugated) in blood. The free BPA might have reached other organs, like kidneys and lungs, where it caused oxidative stress-induced cytoarchitectural damage. Damage to both liver and kidneys might have increased the workload of the heart. The compromised state of organs for metabolism and excretion together with the oxidative stress due to BPA might have led to myocardial injury, as supported by increased CK-MB levels in both of the treated groups. Moreover, the emphysematous changes in the lungs of rats of the treated groups might have led to ineffective gaseous exchange and increased tissue hypoxia. The tissue hypoxia would have further added to the oxidative stress and even increased the workload on the heart, further supplementing myocardial injury.

These results raise certain concerns on the safety of low doses of BPA and widen the research opportunities for correlation of leached BPA with organ disorders and various diseases, including changes at genetic level, in the near future.

Infant feeding bottles which are commonly used to feed the infants leach out BPA, and the present study shows how this leached BPA can be detrimental for the organs. In the present study, it was revealed as how the process of cleaning, sterilizing, and filling these infant bottles by warm milk/water could easily leach out BPA. It is important to mention, here, that a low concentration of BPA leached at lukewarm temperature produced toxic changes in Group 3 rats. If the temperature is raised to around 60–70 °C, the leaching is expected to increase. Hence, when these bottles are filled with warmer milk/water, the chances of leaching increases many-fold. Moreover, the solubility of BPA is expected be greater in milk, due to the chemical nature of BPA. BPA belongs to group of diphenylmethane derivatives and is lipid-soluble. Hence, feeding milk through the plastic bottles to infants further adds to the problem. Once BPA leaches out from plastic bottles, it cannot migrate back even though temperatures are reduced. Hence, when the infant consumes this milk/water later on, the chances of ingestion of leached BPA still prevail.

BPA is already banned in infant feeding bottles in many countries. However, in countries like India, there are feeding bottles available which do not mention information regarding BPA or the type of plastic. Large numbers of people purchase these bottles. Reports from the study conducted by Toxic Links (India) confirm the presence of BPA in the bottles and sippy cups available in the Indian market [50]. People are unaware of the toxicity of BPA and keep using these bottles to feed their infants. It is pertinent to mention here that BPA exposure is reported to be closely linked with childhood obesity, asthma, affects brain development, in fetus [20,21]. Exposure to BPA in infants may occur through multiple ways, like the use of sippy cups, feeding bottles, teethers, plastic toys, and tinned food/beverages [51,52]. Studies have reported the migration of bisphenols and other endocrine disruptors from baby teethers [52]. Exposure to the chemical is supported by studies that report the presence of BPA in the urine of infants [6]. Most of agencies involved in manufacturing these products claim that bisphenols used in their products are very low dose and within safety limits; however, reports from various low-dose BPA exposure studies have shown varied toxic effects [53,54].

The main limitations of the study were that effect of leached water at higher temperatures was not examined, and chemicals other than BPA leached from the plastic bottles were not detected, which might have been the reason behind the greater damage observed in the plastic leached water-treated group.

## 5. Conclusions

In the present study, we could show that BPA is leached out from plastic infant feeding bottles during routine practice of using these bottles. This leached BPA, in concentrations as low as 0.1 µg/mL was capable of producing oxidative stress-induced tissue injury in rats. The novelty of the present study was that the real-life practice of feeding human infants through infant feeding plastic bottles was simulated on rats. The results of the study force us to think about the safety of these bottles. It has been reported earlier that low doses of BPA are safe and not harmful to humans; further studies are implicated to introspect and re-think on the safety of leached low dose BPA in humans. More studies on the actual practice of plastic ware usage and ingestion of leached BPA through dietary sources in humans need to be taken up in the near future, and may unravel some of the myths regarding the safety of plastic wares.

## Figures and Tables

**Figure 1 toxics-08-00034-f001:**
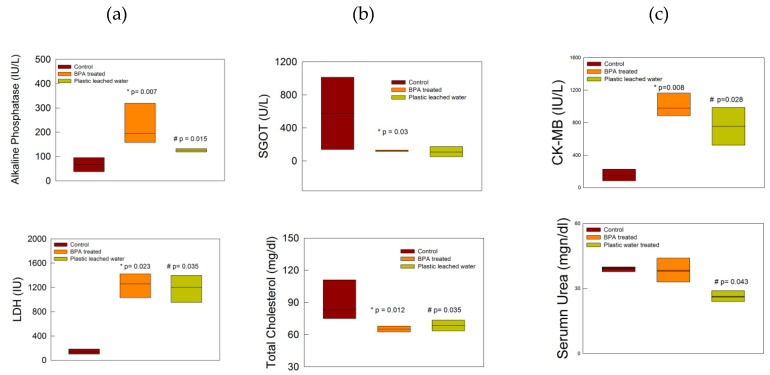
(**a**) Alkaline phosphatase (ALP) and lactate dehydrogenase (LDH) levels increased in the treated groups as compared to control group, (**b**) serum glutamic oxaloacetic transaminase (SGOT) and total cholesterol decreased in treated groups as compared to control group, (**c**) creatine kinase-muscle/brain (CK-MB) markedly increased in both treated groups as compared to control group whereas urea decreased in the plastic leached water-treated group. * and ^#^ indicate *p* < 0.05 Mann–Whitney U test.

**Figure 2 toxics-08-00034-f002:**
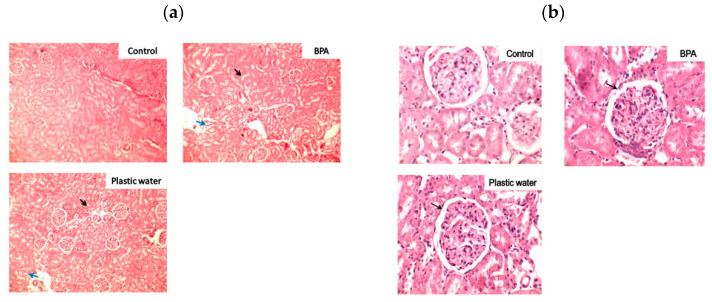
(**a**) Photomicrograph of kidneys of rats magnified 100×. The black arrows show hyperplastic changes in the treated groups and blue arrows indicate tubular thinning. (**b**) Photomicrograph of kidneys of rats magnified 400×. The black arrows show reduced periglomerular space due to hyperplasia in the treated groups. Lymphocytic infiltration is also seen in the treated groups.

**Figure 3 toxics-08-00034-f003:**
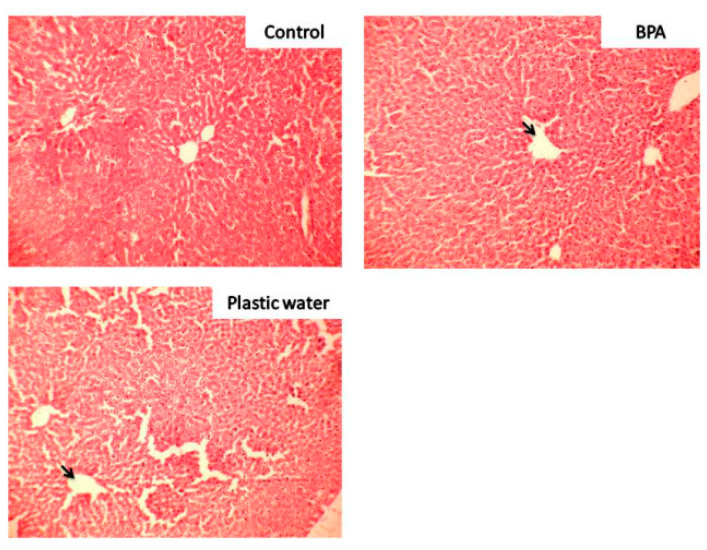
Photomicrograph of liver of rats magnified 100×. The black arrows show dilatation of the central veins in both the treated groups. Loss of normal cytoarchitecture is also seen in both the groups.

**Figure 4 toxics-08-00034-f004:**
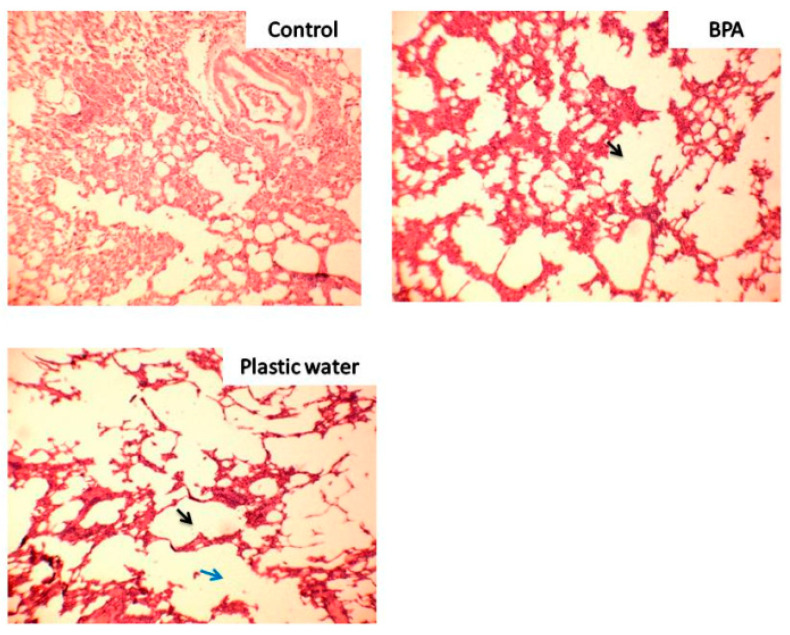
Photomicrograph of lungs of magnified rats 100×. The black arrows show emphysematous changes in the treated groups and blue arrows indicate coalescence of the alveoli.

**Figure 5 toxics-08-00034-f005:**
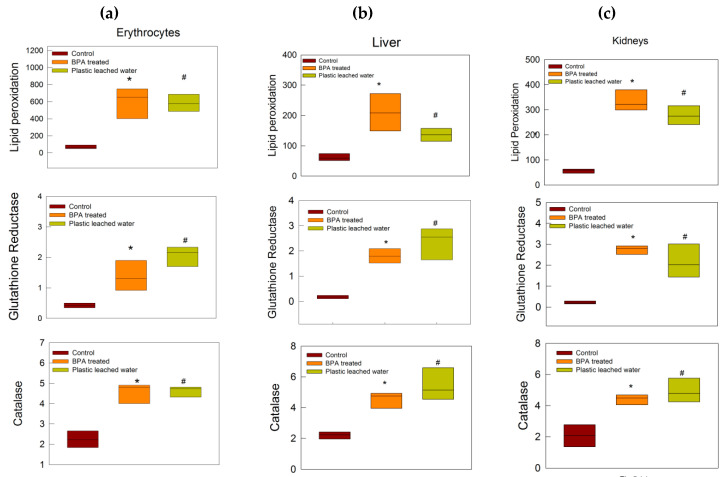
Oxidative stress-related enzyme activity in (**a**) erythrocytes, (**b**) liver, and (**c**) kidneys. Lipid peroxidation, glutathione reductase, and catalase activity increased in both the treated groups as compared to control rats. * and ^#^ indicate *p* < 0.05, Mann–Whitney U test.

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
