# Peer review of "Effect of Exposure of Plastic Infant Feeding Bottle Leached Water on Biochemical, Morphological and Oxidative Stress Parameters in Rats"

_toxics, 2020, doi:10.3390/toxics8020034_

Round 1
Reviewer 1 Report
In this in vivo rat study, the effect of BPA-containing water leached from feeding bottles was compared to direct BPA exposure and control groups. To this end, similar effects on histological and biochemical parameters between direct BPA exposure and baby bottle leached water compared to controls were observed indicating that even small concentrations of BPA might be of concern for human health.
The gained results of the study are valuable for TOXICS readership. Although the manuscript is mainly well-written, the grammar and language style should be improved throughout the manuscript. The manuscript would benefit from an extended discussion evaluating the observed effects regarding the current BPA exposure infants are confronted with.
- Which infant bottles were used for the study? Are these bottles still used by the general population? BPA is banned worldwide from feeding bottles. Are there any studies evaluating the current exposure of infants by BPA via feeding bottles/cups and the associated toxicological effects?
- How is the overall BPA exposure of the plastic leached water group? Was the amount of consumed water recorded? Inline, is there any information on measurable BPA concentration in the tap water used for the control group?
- Are there any other chemicals detected in the plastic leached water that might contribute to the observed adverse health effects?
- How were the biochemical parameters measured? Please, describe the methods in more detail.
- For the statistical evaluation of differences in biochemical parameters between groups, a t-test was applied, whereas for oxidative stress parameters a one-way ANOVA was conducted. What was the rationale behind to perform different tests?
Minor:
- Line 179-181: Please delete the sentence: “This section may be divided by subheadings. It should provide a concise and precise description of the experimental results, their interpretation as well as the experimental conclusions that can be drawn.” Also, in line 205-207 delete the doubled sentence “Photomicrograph of kidneys of rats 400 times magnified. The black arrows show reduced peri-glomerular space due to hyperplasia in the treated groups. Lymphocytic infiltration is also seen in the treated groups.”.
- Line 150: Please indicate centrifugation force in g, not rpm.
- Figure descriptions are confusing. Consider substituting the @ sign in the figures by #, or only *.
- Line 129: Please, add a reference or indicate the respective method section.
- Consider changing (Line 122) “Experimental Protocol“ to “Histological examination and Biochemical parameter”, and moving the first part including the group description at the beginning of the method section.
Author Response
Response to Reviewers’ Comments
The authors are extremely thankful to the reviewers for reviewing the manuscript and providing their valuable suggestions. The authors have tried to modify as per suggestions from learned reviewers and incorporate the changes in revised manuscript. The changes are written in red font in the manuscript “Pant et al MS Revised file”.
General comments:
In this in vivo rat study, the effect of BPA-containing water leached from feeding bottles was compared to direct BPA exposure and control groups. To this end, similar effects on histological and biochemical parameters between direct BPA exposure and baby bottle leached water compared to controls were observed indicating that even small concentrations of BPA might be of concern for human health.
The gained results of the study are valuable for TOXICS readership. Although the manuscript is mainly well-written, the grammar and language style should be improved throughout the manuscript. The manuscript would benefit from an extended discussion evaluating the observed effects regarding the current BPA exposure infants are confronted with.
The authors have tried to improve the grammar and language style in the revised manuscript. Further the discussion section has also been revised.
Page-10; lines-404-409
- Which infant bottles were used for the study? Are these bottles still used by the general population? BPA is banned worldwide from feeding bottles. Are there any studies evaluating the current exposure of infants by BPA via feeding bottles/cups and the associated toxicological effects?
The infant feeding bottles used in the present study were made of plastic. These bottles were purchased from a local shop. The authors declare here that these bottles had no display of type of plastic.
The authors agree that BPA is banned from feeding bottles; however reports from study conducted by Toxic Links (India) confirm the presence of BPA in the bottles and sippy cups available in Indian market. The reference has been cited in the revised manuscript.
Page-3 ; lines-117-119
Page-10; lines-401-403
Ref no.-50
- How is the overall BPA exposure of the plastic leached water group? Was the amount of consumed water recorded? In line, is there any information on measurable BPA concentration in the tap water used for the control group?
The group 3 rats were allowed to drink only plastic leached and not tap water for 30 days. This had exposed these rats to be exposed to the chemical throughout the duration of study.
Each rat drank approximately 30-35 ml of the leached water daily.
The tap water provided to rats in group 1 was RO filtered water. Estimation of BPA concentration in this drinking tap water was not performed.
Page-3; lines- 108-109
- Are there any other chemicals detected in the plastic leached water that might contribute to the observed adverse health effects?
Plastic wares leach out not only BPA but other chemicals like phthalates, BPS, BPF which have estrogenic activity. Even BPA free plastics release chemicals with estrogenic activity. The present study was limited to estimation of BPA only and the authors have not detected the others, which is also the limitation of the present study.
Page- 9; lines- 356-360
Page- 10; lines-412-414
- How were the biochemical parameters measured? Please, describe the methods in more detail.
The authors have now mentioned the details in revised manuscript
Page-4; lines- 153-158
- For the statistical evaluation of differences in biochemical parameters between groups, a t-test was applied, whereas for oxidative stress parameters a one-way ANOVA was conducted. What was the rationale behind to perform different tests?
T-test for unpaired observation was applied both for biochemical parameters and oxidative stress parameters. However one way ANOVA was applied additionally for oxidative stress parameters so as to reduce any chances of error and be more confident about the test results. In both tests, p value was <0.05. The authors however have now used one way- ANOVA for Biochemical parameters also in the revised manuscript to remove any confusion and incorporated the same in revised manuscript.
Page-5; lines- 210-212
Minor:
- Line 179-181: Please delete the sentence: “This section may be divided by subheadings. It should provide a concise and precise description of the experimental results, their interpretation as well as the experimental conclusions that can be drawn.” Also, in line 205-207 delete the doubled sentence “Photomicrograph of kidneys of rats 400 times magnified. The black arrows show reduced peri-glomerular space due to hyperplasia in the treated groups. Lymphocytic infiltration is also seen in the treated groups.”.
The authors have now deleted the lines as suggested by reviewer
- Line 150: Please indicate centrifugation force in g, not rpm.
The authors have indicated the cgf instead rpm
Page-5; line- 197
- Figure descriptions are confusing. Consider substituting the @ sign in the figures by #, or only *.
The authors have substituted @ sign with # in Fig 1 and 5
- Line 129: Please, add a reference or indicate the respective method section.
The authors have incorporated respective method section in revised manuscript
Page-3; line-107
- Consider changing (Line 122) “Experimental Protocol“ to “Histological examination and Biochemical parameter”, and moving the first part including the group description at the beginning of the method section.
The authors have changed the heading and also moved the first part in beginning of method section as suggested by reviewer
Page-4; line-146
Page-3;lines-100-109

Reviewer 2 Report
You must improve the section of material and methods"adult rats of Wistar strain weighing 200-300 grams" female or male? age?
"BPA (analytical grade) was procured from Sigma Aldrich (USA. the rats were ingested with BPA dissolved in olive oil (20 μg/kg".How you have prepared the BPA solution
"The serum thus obtained was used for biochemical analysis to 132 study alkaline phosphatase (ALP), Lactate Dehydrogenase (LDH), serum glutamic-oxaloacetic 133 transaminase (SGOT), Serum glutamic pyruvic transaminase (SGPT), Bilirubin levels, Very-low-134 density lipoprotein (VLDL), high-density lipoproteins (HDL), Total cholesterol, creatine kinase-135 muscle/brain (CK-MB) and Blood Urea levels."You have to give more details
"processed for histopathological examination" more details are needed
Only 6 animals per group? Are the results statistically sufficient? and can you use parametric test? The results do not say anywhere how many samples were analyzed per group.
More details about microscope photos are necesary?
The discussion do not provide sufficient background and not include relevant references.
Author Response
Response to Reviewers’ Comments
The authors are extremely thankful to the reviewers for reviewing the manuscript and providing their valuable suggestions. The authors have tried to modify as per suggestions from learned reviewers and incorporate the changes in revised manuscript. The changes are written in red font in the manuscript “Pant et al MS Revised file”.
You must improve the section of material and methods
- "adult rats of Wistar strain weighing 200-300 grams" female or male? age?
The authors have now revised and mentioned the details in revised manuscript
Page- 2; lines- 89-90
- "BPA (analytical grade) was procured from Sigma Aldrich (USA. the rats were ingested with BPA dissolved in olive oil (20 μg/kg". How you have prepared the BPA solution?
The authors have described the BPA solution preparation in:
Page- 3; lines- 112-114
- "The serum thus obtained was used for biochemical analysis to 132 study alkaline phosphatase (ALP), Lactate Dehydrogenase (LDH), serum glutamic-oxaloacetic 133 transaminase (SGOT), Serum glutamic pyruvic transaminase (SGPT), Bilirubin levels, Very-low-134 density lipoprotein (VLDL), high-density lipoproteins (HDL), Total cholesterol, creatine kinase-135 muscle/brain (CK-MB) and Blood Urea levels."You have to give more details
The authors have mentioned the details in revised manuscript
Page-4; lines-153-158
- "processed for histopathological examination" more details are needed
The authors have now incorporated details of process for histopathological examination in revised manuscript
Page-4; lines-167-187
- Only 6 animals per group? Are the results statistically sufficient? and can you use parametric test? The results do not say anywhere how many samples were analyzed per group.
The animal number per group was calculated using Resource Equation Approach, where a group can have 5-7 animals. Hence we kept 6 animals per group, with total number of rats 18 in all three groups. According to the resource equation approach, T-test and one-way ANOVA can be applied with this sample size.
Ref no. 32: Arifin, W.N.; Zahiruddin, W.M. Sample Size Calculation in Animal Studies Using Resource Equation Approach. Malays J Med Sci 2017, 24(5): 101–105.
The data of all the samples per group were analyzed and pooled to draw the results
Page- 3; lines- 100-101
Page-5;lines- 209-210
- More details about microscope photos are necessary?
The authors have revised the histology description
Page- 6; lines- 245-246, 250-254
Page-7; lines- 264,267-269,274,275,280-282
- The discussion do not provide sufficient background and not include relevant references.
The discussion section has been revised and newer references included in the revised manuscript

Round 2
Reviewer 1 Report
All points have been addressed adequately.
Minor:
line195: What is cgf? Please change to 736 g for 10min.
line 409: Please, add references for:"however reports from various low dose BPA exposure studies have shown varied toxic effects."
Author Response
The authors are extremely thankful to the reviewers for reviewing the manuscript and providing their valuable suggestions. The authors are thankful to reviewer for accepting the responses and revision. The changes are written in red font in the manuscript “Pant et al MS revised MS toxics-777477 ”.
Minor:
- line195: What is cgf? Please change to 736 g for 10min.
The authors have mentioned in grams
Page- 5; line-197
- line 409: Please, add references for:"however reports from various low dose BPA exposure studies have shown varied toxic effects."
The authors have added references as Ref n. 53,54 under reference section
Page-14; lines- 602-606

Reviewer 2 Report
The sample size is not enough.The statistical calculations are not well done. Parametric tests cannot be done with six samples per experimental group
Author Response
The authors are extremely thankful to the reviewers for reviewing the manuscript and providing their valuable suggestions. The authors have tried to modify as per suggestions from learned reviewers and incorporate the changes in revised manuscript. The changes are written in red font in the manuscript “Pant et al MS revised MS toxics- 777477”
- The sample size is not enough. The statistical calculations are not well done. Parametric tests cannot be done with six samples per experimental group
The authors declare that the sample size cannot be increased in the present study, hence non parametric test in the form of Mann Whitney U test has been applied. All the graphs (Fig 1 (a, b,c); Fig 5 (a, b, c) have been redrawn. The result section has been modified accordingly.
Page- 5; lines- 211,212, 218,219, 221,223,228,235
Page-6, lines- 238, 245
Page-8; line- 297

Round 3
Reviewer 2 Report
0